# The Evolution of Diet and Morphology in Insular Lizards: Insights from a Replicated Island Introduction Experiment

**DOI:** 10.3390/ani13111788

**Published:** 2023-05-27

**Authors:** Colin M. Donihue, Anthony Herrel, Maxime Taverne, Johannes Foufopoulos, Panayiotis Pafilis

**Affiliations:** 1Institute at Brown for Environment and Society, Brown University, Providence, RI 02903, USA; 2UMR 7179 Centre National de la Recherche Scientifique/Muséum National d’Histoire Naturelle, Département Adaptations du Vivant, Bâtiment d’Anatomie Comparée, 55 rue Buffon, 75005 Paris, France; anthony.herrel@mnhn.fr (A.H.); maxime.taverne@orange.fr (M.T.); 3Department of Biology, Functional Morphology, University of Antwerp, 2000 Antwerp, Belgium; 4Department of Biology, Evolutionary Morphology of Vertebrates, Ghent University, 9000 Ghent, Belgium; 5School for Environment and Sustainability, University of Michigan, Ann Arbor, MI 48109, USA; jfoufop@umich.edu; 6Department of Biology, National and Kapodistrian University of Athens, 15784 Athens, Greece; ppafil@biol.uoa.gr; 7Zoological Museum, National and Kapodistrian University of Athens, 15784 Athens, Greece

**Keywords:** diet, bite force, *Podarcis erhardii*, Greece

## Abstract

**Simple Summary:**

Small island environments can drive rapid changes in animal traits that provide an advantage for individuals attempting to capitalize on scarce resources. One such trait for lizards is bite force—individuals with stronger bites may be better able to access food that is hard to chew and/or guard access to food and territories from competitors. We introduced lizards to five small, uninhabited islets in Greece and tracked changes in the physical features and bite force of the lizards over several generations. We found changes in body size and bite force that suggested that competition was intense on these small islets; however, we did not see a consistent change in diet among these experimental populations. Our results suggest that competition is an important driver of changes in these traits, and changes in diet may be follow-on consequences. It remains to be discovered if diet is also a driver of these changes in other times of the year when food is especially limited. Some species can and do flexibly and rapidly change in response to changes in their environment.

**Abstract:**

Resource-limited environments may drive the rapid evolution of phenotypic traits and ecological preferences optimizing the exploitation of resources. Very small islands are often characterized by reduced food availability, seasonal fluctuations in resources and strong unpredictability. These features may drive the evolution of phenotypic traits such as high bite forces, allowing animals to exploit a wider variety of the available resources. They may also lead to more generalist dietary patterns in response to food scarcity. However, the lack of predators and competitors on such small islands often also leads to high densities and the evolution of strong sexual dimorphism, which may also drive the evolution of bite force. Here, we take advantage of a unique replicated introduction experiment to test whether lizards introduced into very small islands alter their feeding ecology and use different resources, resulting in the evolution of a large body size, large head size and large bite forces. Our results show that three years after their introduction, the island lizards were larger and had greater bite forces and more pronounced sexual dimorphism. However, the diets were only marginally different between animals from the source population on a very large nearby island and those on the islets. Moreover, distinct differences in diet between animals on the different islets were observed, suggesting that the local environment is a strong driver of resource use. Overall, lizards with absolutely and relatively (adjusted for body size) large bite forces did eat larger and harder prey. Taken together, our data suggest that intraspecific competition is an important driver of the rapid evolution of bite force, which may allow these lizards to exploit the scarce and fluctuating resources on the islets. Whether or not lizards will evolve to include other types of food such as plants in their diet, facilitated by their large bite forces, remains to be explored in future studies.

## 1. Introduction

Food acquisition is a central component of an organism’s life history, as it provides energy and nutrients for survival, reproduction, and dispersal. Thus, the ability to consume a wide variety of food items may provide an organism with a selective advantage. Indeed, generalists, feeding on a vast dietary spectrum, have better chances to survive than specialists under varying ecological conditions or in ecosystems with low food availability [1,2]. Food availability may affect numerous aspects of organisms’ overall biology, such as morphology or ecology [3,4]. In food-limited environments, such as deserts or small islands, the ability to consume hard, large or unusual prey may be at a premium [5,6,7,8,9].

Very small islands and islets are challenging environments often characterized by a reduced abundance and diversity of food resources [7,10]. In many systems, island size is correlated with food availability and the functional diversity of prey resources, causing mesopredators on small islets to switch from their normal diets to alternative food resources, such as plants or marine subsidies [6,11,12,13,14,15,16,17]. In addition, small islets are often extreme environments characterized by periods of food scarcity and strong seasonality [18,19,20] that deviate in many respects from the large-island norms [21,22,23].

Previous studies have shown that on islands, lizards often become larger and have higher bite forces [6]. Consuming hard and large prey items may place selective pressures on the ability to consume a variety of prey and may drive the evolution of head morphology and performance [14]. According to the literature, bite force is a critical performance trait that is linked to the type and size of prey that an organism can consume [14,24,25,26,27,28]. Although some studies have suggested that this is the result of high population densities and strong sexual dimorphism [29], others have suggested that a dietary shift to herbivory [6,13,15] may be an important driver of the pattern of higher bite forces among insular lizard populations. Surprisingly, the evolution of diet has remained rather poorly studied. Consequently, whether or not changes in diet, as observed on islands, take place on ecological timescales remains largely unknown (however, see [15]). Moreover, whether changes in diet precede or follow changes in the ability to cope with food resources is unknown. For example, the ingestion of items that are difficult to reduce, such as plants or seeds, may be a direct driver of larger body size, larger head size and greater bite force, acquired through natural selection [30]. Alternatively, sexual selection through increased density and competition may drive the evolution of high bite forces, which then permit the inclusion of different dietary items in the diet, including large or difficult-to-reduce items [31].

Unfortunately, teasing apart these scenarios is nearly impossible through post-hoc comparisons of populations and instead requires studies following populations over time. Here, we take advantage of a unique experiment in which lizards were introduced from a single-source population on a large island into five very small and geographically close islets with limited food availability (Figure 1). We revisited the islets and the source population three years after the initial introduction, measured the lizards and flushed the stomachs of the lizards to quantify their diets. We predicted that after several generations, the lizards living on the experimental islets would be larger and have stronger bites than lizards on the source island. Accordingly, we predicted that the lizards on the experimental islets would therefore consume larger and more difficult-to-reduce food items. We also expected populations on the different islets to show different diets, as local food availability is likely an important driver of differences in diet [7]. Finally, we predicted that individual lizards with stronger bite forces would eat larger and harder prey, irrespective of the population of origin [32].

## 2. Materials and Methods

*Podarcis erhardii*, the Aegean Wall Lizard (Figure 1), is a medium-sized (50–75 mm adult snout–vent length, [33]) mesopredator that is widely distributed across the Aegean Sea archipelagos and can be found living on the largest islands and even small rocky outcropping islets [34,35]. The species is an invertebrate generalist [33] consuming snails and a wide range of arthropods, including larvae [29,36], while cannibalism has also been reported [37]. In contrast to several other *Podarcis* species, insular *P. erhardii* populations are not known for shifting their diet consistently to plant material [36]. However, frugivory and limited herbivory have previously been observed among small-island populations [29,38].

This study formed part of a multi-island lizard introduction experiment (see [39]). Briefly, during the spring of 2014, we identified five small islets (0.002–0.004 km^2^ surface area) near the islands of Naxos and Paros (Cyclades, Aegean Sea, Greece) that were lacking *Podarcis* lizards (Figure 1) and their terrestrial predators. These islets are arid and composed of a rocky limestone substrate with little soil. Typical of the small Cycladic islands, these islets are patchily covered with plants, particularly *Juniperus* shrubs, small flowering forbs and grasses [40,41]. These islets are used by seabirds as nesting sites [23,42]. The five islets used in this experiment were selected for their comparable vegetation communities and nearly identical weather patterns (given their close proximity to each other). The differences that do exist between these islets in vegetation cover, island size and topography are vastly overshadowed by the difference between the five islets and the source island of Naxos.

We introduced 8 males and 12 females from an abundant large-island population on Naxos (430 km^2^ surface area) into each of the five experimental islets. In the ensuing years, the populations were surveyed annually. In the spring of 2017, we sampled the experimental introduction populations along with the source population on Naxos by catching lizards by hand and assessing their body size, bite force and diet. The lizards sampled in 2017 were descendants of the original introduction lizards—none of the individually marked introduction lizards were found in 2017. In every survey, morphological measurements were taken by the same researcher (CMD) using a digital caliper (Mitutoyo 500–752), and bite force measurements were taken by a single researcher (AH) using a purpose-built bite force meter (Kistler 9203, ±500 N connected to a Kistler 5995A charge amplifier). See [31] for a complete description. In total, we collected morphology, performance and diet data from 194 lizards.

Shortly after capture, each lizards’ stomach was flushed using a ball-tipped gavage needle and water following standard protocols [29,32], and the stomach contents were transferred to tubes with a 70% ethanol solution. In the lab, the stomach contents were identified, and the length, width and volume of each food item was estimated (Figure 1). We classified the diet items into prey categories according to their hardness and their evasiveness (soft, medium, hard; sedentary, intermediate, evasive), following [43,44].

### Statistical Methods

All analyses were conducted with R version 4.2.0 (R Core Team, 2022). Before analysis, we log_10_-tranformed all the prey characteristics to follow assumptions of normality for the analyses. Additionally, we arcsine-transformed the calculated proportions of different items in the diet. We calculated the body condition of the lizards using the scaled mass index (SMI), following [45]. When testing for differences between the experimental islet populations and source population, we used experimental islet identity as a random effect. Accordingly, we evaluated our models using the ‘*glmer*’ function in the ‘*lme4’* R package [46] and investigated the significance of the factors using Type III Analysis of Variance in the ‘*car*’ package [47]. When testing for differences in diet and bite force within the experimental islets, island identity was tested as a fixed effect, and the models were again evaluated using the ‘*lme4’* R package [46]. All figures were created using the ggplot2 package in the tidyverse [48].

## 3. Results

### 3.1. Comparing the Morphology and Bite Force of Lizards from the Experimental Populations to those of the Source Population on Naxos

In 2017, lizards on the introduction islets were significantly larger (Table 1) than those measured from the source population on Naxos (*F*_1_ = 13.14, *p* = 0.0207; Figure 2). We did not find a difference in body condition between the treatment and control populations (*F*_1_ = 0.10, *p* = 0.7592); however, males tended to have a higher SMI than females (*F*_1_ = 144.37, *p* < 0.0001; Figure 2). When testing for differences in body size, the lizards on the experimental islets had a stronger maximum bite force (*F*_1_ = 712.56, *p* < 0.0001). We detected a significant interaction between sex and location: the differences in bite force between the introduction islets and the source population were more substantial in males than in females (interaction: *F*_1_ = 20.88, *p* < 0.0001), resulting in more pronounced sexual dimorphism on the islets (Figure 2).

### 3.2. Comparing Morphology and Bite Force between the Five Experimental Populations

While the introduction propagules in 2014 were all drawn from the same population and were statistically indistinguishable, in 2017, we found significant differences in the body size (*F*_4_ = 4.83, *p* = 0.0009), body condition (*F*_4_ = 4.206, *p* = 0.0025) and bite performance (*F*_4_ = 13.068, *p* < 0.0001) of the islet populations (Table 1). We found that, on average, the SVL of lizards on Mavronissi was the greatest, while the lizards on Kambana were the smallest. In contrast, the Kambana lizards had the highest body condition index, while the Petalida lizards had the lowest body condition (Table 1). Galiatsos and Kambana lizards both had the hardest bites for their body size, while Agios Artemios and Mavronissi lizards had relatively weak bites.

### 3.3. Comparing the Diet of Lizards from the Experimental Populations to That of the Source Population on Naxos

Generally, we did not find a marked difference in the min, max or mean prey dimensions between the introduction and source populations (all *p* > 0.05). This is largely because the variance in the size of prey ingested by the source population lizards on Naxos was high, encompassing the variance in prey size found on the islets. We did detect a significant interaction between sex and treatment: on Naxos, females consumed significantly longer and thus larger-volume prey than males (*F*_1_ = 4.150, *p* = 0.0424). This difference was due to a high proportion of female lizards that had ingested larvae.

Comparing the average hardness [43] of the diet items between the source and introduction islet populations, we again found a significant interaction with sex. While males and females on the introduction islets ate comparably hard diet items relative to males on Naxos, the diet of the Naxos females significantly differed from that of the experimental females and males, being primarily composed of soft prey items (*F*_1_ = 16.133, *p* < 0.0001). Again, this trend is explained by the fact that females on Naxos ate a high proportion of soft-bodied larvae.

We also compared the number of hard and soft taxa in the diets of the lizards. On the introduction islets, males consumed a higher proportion of hard prey items than females (*F*_1_ = 5.005, *p* = 0.0252; Figure 3). We found that the Naxos population generally consumed more hard taxa than the introduction islet populations (*F*_1_ = 8.524, *p* = 0.0233). The pattern was reversed for the number of soft taxa, but while the sex differences were maintained (*F*_1_ = 5.214, *p* = 0.0230), the differences between the source and treatment islet populations was only marginally significant (*F*_1_ = 4.327, *p* = 0.0852).

Comparing the escape abilities of the prey taxa, we found that females generally consumed a higher number of sedentary prey items (*F*_1_ = 14.688, *p* = 0.0002), while males consumed significantly more evasive prey items (*F*_1_ = 7.884, *p* = 0.0053). We also found significant differences between the treatments, with lizards on the introduction islands consuming significantly more sedentary taxa (*F*_1_ = 6.541, *p* = 0.0422) and significantly fewer intermediate taxa (*F*_1_ = 8.283, *p* = 0.0240). We found no difference in the number of evasive prey items consumed between the introduction islets and source island (*F*_1_ = 0.481, *p* = 0.5180; Figure 3).

The number of animals with plant material in their stomachs differed significantly between the treatment and control islands. Plant consumption, measured as the number of plant items, was higher among the source population than the introduction populations (*F*_1_ = 7.346, *p* = 0.0460; Figure 3). Moreover, we found a significant interaction: males on Naxos consumed more plants than females (*F*_1_ = 4.787, *p* = 0.0294). The total volume of plant material in the stomachs of the Naxos lizards was also higher than that of the introduction islet animals (*F*_1_ = 26.022, *p* = 0.0016).

### 3.4. Comparing Diet Composition between the Five Experimental Populations

We found no significant differences in the mean length or width of the prey items consumed between the sexes or between islet populations. We did, however, detect a significant difference in the mean volume of prey between the islets and between the sexes. While, alone, there was no sex difference (*F*_1_ = 0.219, *p* = 0.6398), the significant sex by islet interaction (*F*_4_ = 2.830, *p* = 0.0249) was most prominent in the Agios Artemios lizards, with females consuming more smaller-volume prey compared to males (Figure 4A).

With respect to the mean hardness [43] of the prey items consumed on the islands, we did not find a difference between the sexes, but we did find a significant difference between islets (*F*_4_ = 2.830, *p* = 0.0249; Figure 4B). On average, the Kambana lizards’ diet was composed of harder prey items.

Comparing the numbers of soft, intermediate and hard diet items, we found a significant difference between the sexes, with female lizards tending to eat more soft prey taxa (*F*_1_ = 6.026, *p* = 0.0146), and a significant interaction between the sexes and island identity (*F*_4_ = 4.141, *p* = 0.0028), with sex difference being most prominent on Mavronissi. For prey of intermediate hardness, we found differences between islands (*F*_4_ = 2.981, *p* = 0.0194) but not between the sexes. Finally, for the number of hard prey taxa, we again found differences between the sexes and islands. Males tended to eat harder prey items than females (*F*_1_ = 6.879, *p* = 0.0091; Figure 4B), particularly on Kambana, though the reverse was found for the Mavronissi lizards. Similarly, Kambana lizards had a significantly higher proportion of plant material in their diets than the other lizards (*F*_4_ = 10.485, *p* < 0.0001; Figure 4D).

Comparing the proportions of prey of different evasiveness levels, we found that females ingested significantly more sedentary prey (*F*_1_ = 15.053, *p* = 0.0001) than males across the islets (no interaction), and there were significant differences between the islets (*F*_4_ = 3.074, *p* = 0.0166), with Mavronissi lizards eating the most sedentary prey items and Kambana lizards eating the fewest. There was no difference between the sexes in the number of intermediate-activity prey items. However, there were significant differences between the islands (*F*_4_ = 2.650, *p* = 0.0334), with Kambana lizards eating the highest number of intermediate-activity prey items. We found that across the islands, males consumed a higher number of evasive prey taxa (*F*_1_ = 11.026, *p* = 0.0010; Figure 4C), with significant differences between islands (*F*_4_ = 4.844, *p* = 0.0008) and Petalida lizards eating the most.

### 3.5. Testing for the Morphological and Performance Determinants of Diet Characteristics

We tested our hypothesis that individuals with harder bites would tend to eat harder diet items. Indeed, lizards with hard bites did ingest a higher number of hard taxa (*F*_1_ = 5.308, *p* = 0.0219), and lizards with relatively hard bites for their body size also tended to eat more hard taxa (*F*_1_ = 10.001, *p* = 0.0017; Figure 5A). We also tested our hypothesis that hard-biting lizards would tend to eat more evasive prey and, indeed, found that the data supported this prediction. Lizards with stronger bites (*F*_1_ = 6.064, *p* = 0.0143) and strong bites for their body size (*F*_1_ = 12.111, *p* = 0.0006; Figure 5B) ate a higher number of evasive prey taxa. Third, we tested the hypothesis that lizards with absolutely stronger bites would ingest more plant material. Kambana was the only island with a large proportion of herbivory lizards, thus restricting the analysis to that island (*n* = 107). We found a marginally significant (*F*_1_ = 3.137, *p* = 0.0794; Figure 5C) positive relationship. Finally, we tested whether there were differences in diet based upon body size. Specifically, we hypothesized that smaller lizards would consume a higher number of sedentary and soft diet items. Neither relationship was borne out in the data—we found no correlations between body size and diet characteristics.

## 4. Discussion

### 4.1. Differences in Body Size and Bite Force

Lizards on the experimental islets were larger than those found in the source population. The particular conditions leading to gigantism, as predicted by the island rule, are further intensified on small islets [49,50]. This is a fairly consistent pattern among *Podarcis* populations living on Mediterranean islets [10,23,51,52,53,54]. A recent study focusing on *P. erhardii* demonstrated that this species attains its largest body sizes on islands with more resources, whether these are lush vegetation or dense seabird colonies [55]. Furthermore, large body sizes are associated with dense island populations, implicating a role of intraspecific competition. While, at this point, the relative importance of food availability versus competition cannot be determined, it is possible that both processes are responsible for the observed shifts in body size.

Nonetheless, the islet lizards showed no differences in body condition compared to the lizards from the source population. However, overall, lizards in the introduced populations had relatively stronger bite forces irrespective of variation in body size. The observed differences between the source population on Naxos and the introduced populations on the islets were strongest for males. This is suggestive of male–male competition as a driver of the observed differences in bite force. Interestingly, this resulted in greater bite force dimorphism among animals from the introduced populations. The lizards from the different islets differed in body size, body condition and bite force, suggesting strong effects of the local environment. We hypothesize that islet particularities may be the baseline drivers of these patterns. Indeed, idiosyncratic features may alter aspects of lizard biology on small Mediterranean islets [15,56,57].

Hence, it is possible that differences in nesting seabird density and vegetation cover result in differences in the abundance and diversity of arthropods, which, in turn, can impact growth rates, survival and reproductive success [15,17,55,58]. Ultimately, such demographic factors can affect population density, ultimately resulting in greater intraspecific competition. Indeed, *Podarcis* lizards that live on densely populated islets compete fiercely for access to food and mates, and these intraspecific aggressions often result in limb amputations or even cannibalism [29,59,60,61,62]. Given that such patterns are strongest for males, male–male competition is likely an important proximate driver, providing fitness benefits to males with high bite forces and resulting in the rapid evolution of bite force in these populations. Consistent with this scenario, three years after the initial introduction, the densities were highest on Kambana and Galiatsos, two of the populations with particularly strong size-corrected bite forces.

### 4.2. Differences in Diet

Overall, our data showed that diet was quite variable among all the populations sampled. No differences in the size of the prey eaten were detected between the source and introduction islets. However, lizards from the source population appeared to eat more hard prey on average. Moreover, contrary to what might have been expected, animals on the islets ate more sedentary taxa. When comparing the source and the introduced populations, it became apparent that females on Naxos ate more soft, large prey, driving some of the overall differences detected. This is probably related to the availability of insect larvae on Naxos, characterized by more complex vegetation types and a greater diversity of insects. In contrast, the islets are characterized by very low shrubby or grassy vegetation, with few or no bushes. Consequently, the diversity of prey available on the islets is almost certainly lower than that on Naxos.

While the five islets have very similar ecologies, especially relative to the source population, they do differ from each other in a variety of subtle ways, resulting in slightly different environment and arthropod communities. We believe that this explains why three years after the introduction, the lizard diets showed some differences between the sites. For example, although lizards on different islets did not significantly differ in the size of the prey eaten, Kambana lizards tended to eat harder prey on average. Kambana’s vegetation is relatively diverse and complex for an islet, which may be one cause of this difference. As another example, while male lizards generally ate larger prey and a larger number of hard prey than females, this pattern was especially strong on Agios Artemios, one of the islands with a particularly high lizard density. These examples suggest the importance of the immediate environment in determining diet.

### 4.3. Relationships between Diet, Size, and Bite Force

Our results showed that individuals with absolutely and relatively higher bite forces consumed harder and more evasive prey on average. This finding is in accordance with previous studies showing that animals with higher bite forces tend to eat larger and harder prey [32]. The relationship between bite force and the proportion of evasive prey may, at first, seem somewhat counterintuitive but can be explained by examining the types of evasive prey consumed. Many of the evasive prey consumed on the islets were hymenopterans (e.g., bees, wasps), known to be both evasive and hard prey [44]. Given that these islets are situated relatively close to the larger islands of Paros and Antiparos, large flying prey can easily cross over to the islands to take advantage of the flowering plants in spring, when the study was conducted. Interestingly, no relationships between body size and prey size or between body size and the functional properties of the prey were observed. This stands against the often-cited relationship between overall lizard size and the inclusion of large food items, such as plants or even small vertebrate prey, in the diet [25,63,64,65] and suggests that for these small lizards, bite force is the key driver providing access to hard food items [8].

We found highly significant differences in body size and bite force between males and females, with males on the islands having substantially stronger bite forces. Previous studies have suggested that in *Anolis* lizards, sexual dimorphism is largely mediated by plastic differences in growth rate, with males attaining larger sizes in environments with higher food availability [66]. These differences in size and shape have been proposed to enable niche divergence between the sexes, with males and females specializing in different food resources [30,67]. Here, too, the differences in size and bite force gave rise to significant differences in diet between the sexes, with males eating more large, hard and evasive prey. Females, on the other hand, ate more soft and sedentary prey, resulting in reduced overlap in diet between the sexes.

### 4.4. Plant Consumption

Lizards from the source population consumed more plants on average, suggesting that after three years, no significant shifts in diet towards omnivory had taken place. The changes in diet observed in other study systems [15] may have been equally dependent on initial changes in morphology and performance, enabling access to novel resources [58]. On the other hand, these populations might have had more time at their disposal to shift their dietary preferences towards (partial) herbivory. In addition, the vegetation on the Cyclades contains very high percentages of aromatic or otherwise chemically defended taxa, making many of the islet plants unattractive targets for lizard consumption. Overall, males of the introduced populations consumed more plants than females, suggesting that large bite forces may enable the exploitation of these hard-to-digest prey. Of the experimental islets, Kambana lizards ate more plant material than lizards from the other islets. Kambana lizards were in a better body condition and had relatively higher bite forces for their size, which may have further facilitated the inclusion of plant matter in their diet. Consistent with these observations, there was a tendency for lizards with higher bite forces to eat more plants on the island of Kambana. The latter hosts the most diverse and complex vegetation of all the islets. This higher plant abundance may have also played a role in the more frequent consumption of plant material [36,68]. Herbivory represents a consistent trend among islet *Podarcis* lizards [15,19,69,70]. Future follow-up studies of diet and morphology will be of interest to determine whether a more general shift towards herbivory can be observed on the island of Kambana.

## 5. Conclusions

Overall, our results suggest that the introduction of lizards from a large island into small islets resulted in rapid shifts in body size and bite force. However, differences in diet between the source and introduced populations were relatively minor and not in-line with the role of diet as the principal driver of the observed differences in morphology and performance. Moreover, differences in body size and bite force were most notable for males, resulting in stronger sexual dimorphism on the islets. Overall, our results suggest that intraspecific competition related to increased density, in turn associated with a lack of predation, is the principal driver of the observed patterns. However, the observed changes in morphology and bite force also appear to allow animals to eat harder prey and promote plant consumption. As such, sexual selection appears to be an important driver of divergence in morphology and performance, facilitating access to novel resources in these resource-limited environments.

## Figures and Tables

**Figure 1 animals-13-01788-f001:**
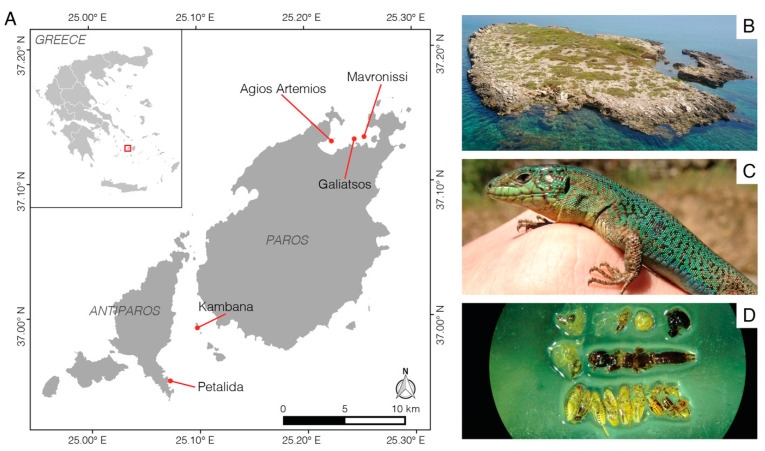
The five experimental islets (**A**) used in this study are situated around the large island of Paros (Cyclades, Aegean Sea, Greece). The study islets (e.g., (**B**)—Agios Artemios), are rocky, arid and covered with a mix of grasses, forbs and low bushes. In 2014, *Podarcis erhardii* lizards (**C**) were introduced into each islet. In 2017, the populations were revisited, and the stomachs of the lizards from each islet were flushed (**D**). The compositions of their diets were then sorted and identified manually. Photo credit (**D**): Kathryn Culhane.

**Figure 2 animals-13-01788-f002:**
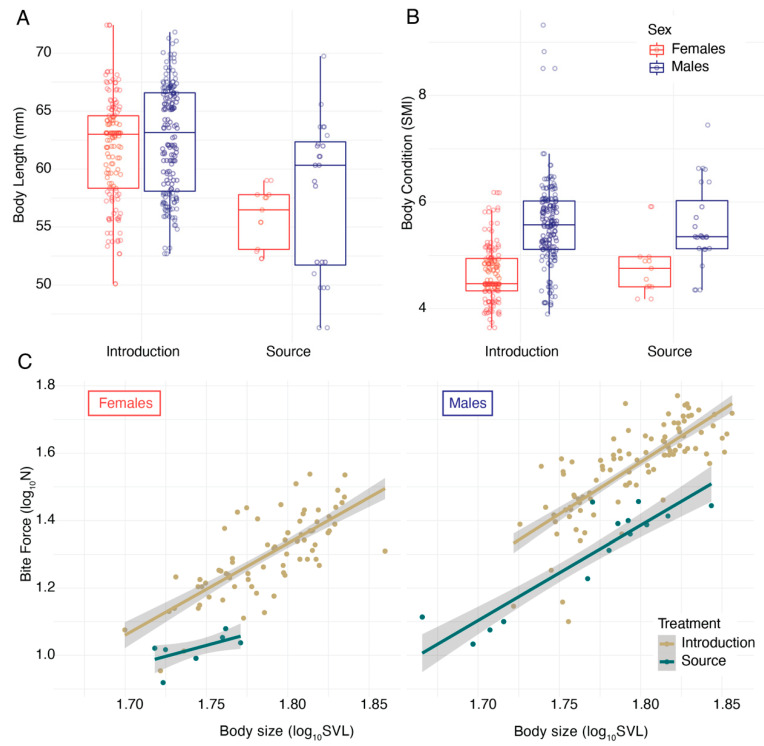
Comparison of the body length (**A**) and body condition (**B**) of lizards from the experimental introduction and control source populations revealed significant differences. Lizards on the introduction islets were significantly longer than lizards from the source population (**A**), and we found significant differences in body condition (**B**) between the sexes (red: females; blue: males) in this experiment. Lizards on the introduction islets tended to be larger, and males tended to have a higher scaled mass index [45]. We also found (**C**) that both female (left) and male (right) lizards on the experimental introduction islets (yellow) had significantly stronger bite forces than lizards from the source population (green) on Naxos.

**Figure 3 animals-13-01788-f003:**
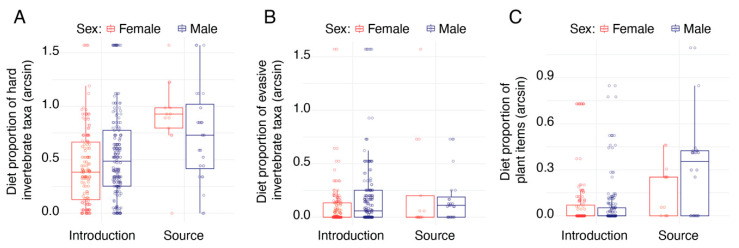
We found that males tended to consume a higher proportion of hard invertebrate taxa than females (**A**) and that, on average, the lizards from the source population consumed more hard taxa than the lizards on the experimental introduction islets. We found that overall, males tended to eat more evasive prey taxa (**B**) than females, but there was no significant difference between the populations on the introduction and source islands. Finally, we found that the lizards from the source population had a higher proportion of plant material than lizards on the introduction islets (**C**), though there was substantial variability among both populations. All proportions were arcsine-transformed for analysis and visualization.

**Figure 4 animals-13-01788-f004:**
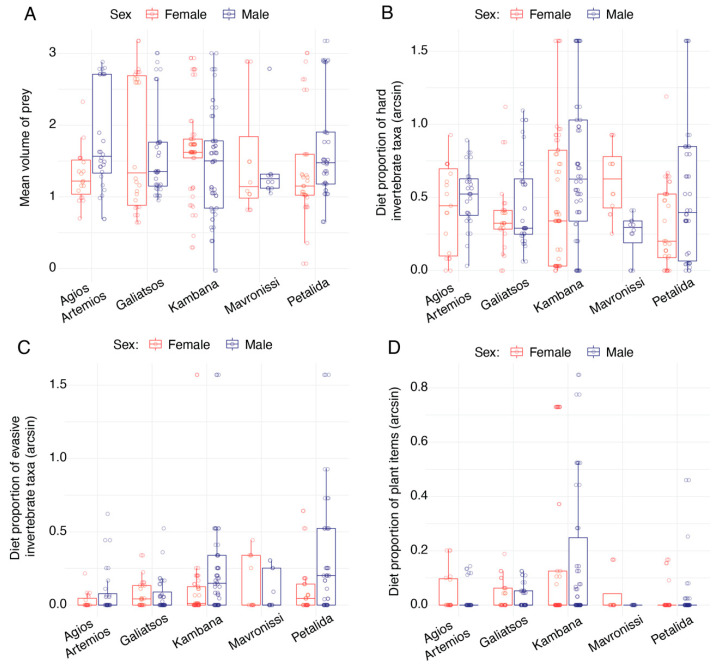
Comparing the diet characteristics of the five experimental populations, we found a significant interaction between sex and island (**A**) in the average volume of prey items. Males (blue) often consumed more larger-volume prey items than females (red). We also found that males tended to eat more harder prey items (**B**) and evasive taxa (**C**) than females. Finally, we found significant differences in the proportion of plant material in the diets (**D**), with lizards on the islet of Kambana having consumed the most plant material. All proportions were arcsine-transformed for analysis and visualization.

**Figure 5 animals-13-01788-f005:**
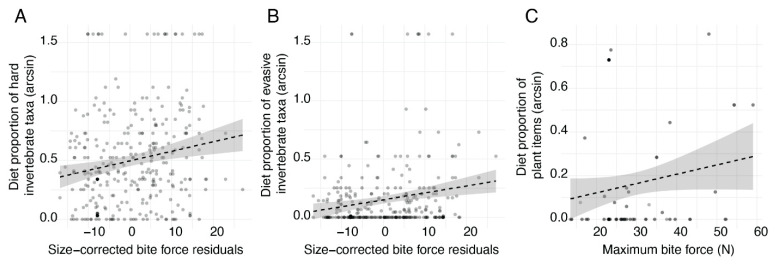
We tested whether bite force was related to differences in diet characteristics and found that lizards with harder bites tended to consume a higher proportion of hard invertebrate taxa (**A**) and evasive invertebrate taxa (**B**), as determined by significant positive relationships. We also tested whether lizards with harder bites tended to consume more plant material on the only island in which sufficient plant material was consumed (Kambana, *n* = 107) and detected a marginally significant positive relationship (**C**). All proportions were arcsine-transformed for analysis and visualization.

**Table 1 animals-13-01788-t001:** Summaries of the body size (SVL), body condition (SMI) and absolute bite force for the male and female lizards on each of the introduction islets as compared to the source population on Naxos (bottom, in gray).

Islets	N		Mean SVL ± SE		Mean Body Condition Index ± SE		Mean Bite Force ± SE	
			(mm)				(N)	
	*F*	*M*	*F*	*M*	*F*	*M*	*F*	*M*
Agios Artemios	19	28	63.8 ± 0.91	63.9 ± 0.95	4.73 ± 0.13	5.37 ± 0.18	20.8 ± 0.83	35.1 ± 1.96
Galiatsos	26	36	60.9 ± 1.14	62.8 ± 0.73	4.47 ± 0.10	5.67 ± 0.15	21.5 ± 1.34	40.0 ± 1.83
Kambana	53	54	61.7 ± 0.54	60.6 ± 0.65	4.70 ± 0.59	5.94 ± 0.09	22.8 ± 0.64	35.9 ± 1.48
Mavronissi	12	12	64.8 ± 1.36	64.5 ± 1.14	4.59 ± 0.13	5.46 ± 0.13	18.4 ± 0.64	40.4 ± 1.62
Petalida	33	42	60.3 ± 0.66	63.6 ± 0.61	4.75 ± 0.12	5.15 ± 0.11	18.0 ± 0.94	39.1 ± 1.23
Naxos: Alyko	13	24	55.9 ± 0.70	57.7 ± 1.36	4.80 ± 0.16	5.54 ± 0.16	9.82 ± 0.86	19.7 ± 1.34

## Data Availability

All data in this study are available upon request from the authors and will be made publicly available pending publication of additional manuscripts.

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
