# Peer review of "The Evolution of Diet and Morphology in Insular Lizards: Insights from a Replicated Island Introduction Experiment"

_animals, 2023, doi:10.3390/ani13111788_

Round 1

Reviewer 1 Report

Review comments

In this study, the authors take advantage of a unique replicated introduction experiment to test whether the Aegean Wall lizards introduced on very small islands alter their feeding ecology and use different resources resulting in the evolution of large body size, large head size and large bite forces. This is an interesting topic that may has important implications for understanding the evolution of diet and morphology in insular lizards. Overall, the manuscript is well written and easy to follow.

However, several issues need to be clarified before the manuscript can be considered for publication (L=line).

L101-112. Reduced predation is also assumed to cause the body size shift of insular species. Did the five small islets have any predators of the Aegean Wall lizards? If not, it should be clearly stated.

L113-122. These five small islets are very small in size (0.002 - 0.004 km2 surface area). Are the 194 lizards you collected all the individuals for all the five small islets? Population size can be used to measure intraspecific competition, which is also important for body size shift of insular species. You can also analyze whether population size affect your results.

L142-146. The description of Statistical methods is relatively simple. It is not clear how you compare the diet and morphology of lizards between the experimental islets and source population and among the experimental islets. Why did you use the ‘glmer’ function and a Type III Analysis of Variance, rather than one-way ANOVA? More details are needed.

L190 and 244. “In comparing the average hardness (Herrel & O'Reilly 2006)”. It is the result section, so (Herrel & O'Reilly 2006) should be deleted.

L352. “4.3. Relationships diet, size, and bite force”, it should be “Relationships between diet, size, and bite force”?

References. Please use the abbreviated or full journal names consistently.

This is an interesting study and I am happy to see it published.

Author Response

Reviewer 1:

In this study, the authors take advantage of a unique replicated introduction experiment to test whether the Aegean Wall lizards introduced on very small islands alter their feeding ecology and use different resources resulting in the evolution of large body size, large head size and large bite forces. This is an interesting topic that may has important implications for understanding the evolution of diet and morphology in insular lizards. Overall, the manuscript is well written and easy to follow.

However, several issues need to be clarified before the manuscript can be considered for publication (L=line).

L101-112. Reduced predation is also assumed to cause the body size shift of insular species. Did the five small islets have any predators of the Aegean Wall lizards? If not, it should be clearly stated.

L101-112: We thank the reviewer for this suggestion. We have updated the text to specifically clarify that all of the experimental islets also lacked predators of the Podarcis lizards.

L113-122. These five small islets are very small in size (0.002 - 0.004 km2 surface area). Are the 194 lizards you collected all the individuals for all the five small islets? Population size can be used to measure intraspecific competition, which is also important for body size shift of insular species. You can also analyze whether population size affect your results.

L113-122. The 194 lizards measured in this study reflect a sample of all of the lizards on the islets. To minimize our effect on the delicate vegetation of these islets, and the large number of lizards, a full census survey was impossible. We agree that tracking population density is a very interesting question for future studies.

L142-146. The description of Statistical methods is relatively simple. It is not clear how you compare the diet and morphology of lizards between the experimental islets and source population and among the experimental islets. Why did you use the ‘glmer’ function and a Type III Analysis of Variance, rather than one-way ANOVA? More details are needed.

L142-146. We thank the reviewer for this feedback. We have taken efforts to expand our description of our statistical approaches. To answer here, we decided a priori to approach this analysis using a mixed effects model methods enabling the use of random effect variables to account for variation between individuals and populations that were not reflected in our hypothesis testing but nonetheless reflect important biological variation. For consistency, we used this same approach throughout the manuscript, even though, as the reviewer correctly points out, some analyses could also have been conducted using one-way ANOVAs. The patterns and differences reported are robust to either approach.

L190 and 244. “In comparing the average hardness (Herrel & O'Reilly 2006)”. It is the result section, so (Herrel & O'Reilly 2006) should be deleted.

L198 and 244: We appreciate the reviewer’s opinion on this matter; however we would prefer to retain the reference in these two instances in the results section. “Average hardness” is a calculated index, not a measured data point in this manuscript, and we would like to emphasize this to the readers by pointing them to the reference where the index was first described.

L352. “4.3. Relationships diet, size, and bite force”, it should be “Relationships between diet, size, and bite force”?

L352: Yes, thank you very much for catching this mistake.

References. Please use the abbreviated or full journal names consistently.

Done

This is an interesting study and I am happy to see it published.

We sincerely appreciate the reviewer's contributions and enthusiastic support.

Reviewer 2 Report

The article entitled: “The Evolution of Diet and Morphology in Insular Lizards: Insights from a Replicated Island Introduction Experiment” is an interesting and well-designed study in which authors explored differentiation in diet and functional and performance traits among populations of lizards from the same origin but developing in differentiated habitats. In my opinion, it is very well-written, the design is appropriate to address the outlined objectives, and data collection and analysis were thoroughly addressed. 

I think this study is appropriate for publication in Animals, although I have some general comments, mostly doubts.

My main concern is about the objectives and the results presentation. First, I miss predictions for some questions addressed later in the manuscript, as for example a prediction regarding body size and body conditions. Also, due to the number of results, I think that reading can be facilitated by exposing the predictions/objectives and corresponding results in the same order. For instance, the results corresponding to the first hypothesis (lines 83-84) are presented at the end of the results (section 3.5). In addition, I think that structuring the results section according to the groups that are compared could be of help for the readership. For example, I suggest merging sections 3.1. and 3.3. (main island vs. islets) and sections 3.2. and 3.4. (differentiation across islets). Or al least, changing the order of the sections to group the results for each grouping (source vs. introductions AND between introductions).

Then, if I understood well, a total of 20 individuals were introduced in each islet from the main one, where the population is large (and I guess, variable). I am not aware on the generation time of the studied species, but it could be possible that any of the sampled individuals in this study was a recapture from the original introduction? Also, can the observed differences between islets-source be the result of founder effects?. In the same line, I was wondering if it possible that observed differences result from plastic changes due to the exposure of differential developmental environment (e.g. available preys). After all, three years is a very short time-spam.   

L352: You state that “Our results showed that individuals with absolute and relatively higher bite forces consumed on average harder and more evasive prey.”. However, despite having weaker bite force than lizards from the islets, Naxos individuals consume more hard preys (and plants) than those from the smaller island (L208-209. How do you explain this pattern? (OK… there it is L403-414).

Some inor comments:

L63-66: I think it is not clear to what “observed patterns” this sentence refers to. I understand that it refers to higher bite forces, but it could be useful explicitly stating it here for clarity.

L139-140: homoscedasticity assumption?

L144: Due to the high number of analyses you did, it could be useful listing the response variables tested in the models.

L165: it could be informative adding to the figure what pair differ significantly (Figure 2B).

Author Response

Reviewer 2:

The article entitled: “The Evolution of Diet and Morphology in Insular Lizards: Insights from a Replicated Island Introduction Experiment” is an interesting and well-designed study in which authors explored differentiation in diet and functional and performance traits among populations of lizards from the same origin but developing in differentiated habitats. In my opinion, it is very well-written, the design is appropriate to address the outlined objectives, and data collection and analysis were thoroughly addressed. 

I think this study is appropriate for publication in Animals, although I have some general comments, mostly doubts.

My main concern is about the objectives and the results presentation. First, I miss predictions for some questions addressed later in the manuscript, as for example a prediction regarding body size and body conditions. Also, due to the number of results, I think that reading can be facilitated by exposing the predictions/objectives and corresponding results in the same order. For instance, the results corresponding to the first hypothesis (lines 83-84) are presented at the end of the results (section 3.5). In addition, I think that structuring the results section according to the groups that are compared could be of help for the readership. For example, I suggest merging sections 3.1. and 3.3. (main island vs. islets) and sections 3.2. and 3.4. (differentiation across islets). Or al least, changing the order of the sections to group the results for each grouping (source vs. introductions AND between introductions).

We thank the reviewer for this helpful feedback. We have now changed the last paragraph of the introduction to include a prediction on changes in morphology and bite force and we have changed the wording of the section headers in the Results section to be clearer about the progression of comparisons being made.

Then, if I understood well, a total of 20 individuals were introduced in each islet from the main one, where the population is large (and I guess, variable). I am not aware on the generation time of the studied species, but it could be possible that any of the sampled individuals in this study was a recapture from the original introduction? Also, can the observed differences between islets-source be the result of founder effects?. In the same line, I was wondering if it possible that observed differences result from plastic changes due to the exposure of differential developmental environment (e.g. available preys). After all, three years is a very short time-spam.   

The reviewer makes a very good point and we will clarify in the text that none of the lizards sampled in 2017 were among the initial 20 introduced to the islets. We can rule out some degree of founder effects – the 20 individuals introduced to each islet were selected at random from a large sample on Naxos and were statistically indistinguishable in morphology and performance in 2014 when they were introduced. Whether there were founder effects in behavior or developmental plasticity potential is unknown.  

L352: You state that “Our results showed that individuals with absolute and relatively higher bite forces consumed on average harder and more evasive prey.”. However, despite having weaker bite force than lizards from the islets, Naxos individuals consume more hard preys (and plants) than those from the smaller island (L208-209. How do you explain this pattern? (OK… there it is L403-414).

L352: Yes, we thank the reviewer – our best explanation is in L403-414.

Some minor comments:

L63-66: I think it is not clear to what “observed patterns” this sentence refers to. I understand that it refers to higher bite forces, but it could be useful explicitly stating it here for clarity.

L63-66: We have clarified this to specify we were talking about the observed pattern in bite force among insular populations.

L139-140: homoscedasticity assumption?

L139-140: Following the data transformations, the data met the assumptions of normality and homoscedasticity.

L144: Due to the high number of analyses you did, it could be useful listing the response variables tested in the models.

L144: We appreciate the reviewer’s suggestion. We attempted writing out a description of the response variables tested in the models but felt it was ultimately more confusing and difficult to follow for readers. We also attempted to convey the information in a table, but felt that it didn’t provide much insight unless each model was more thoroughly described and results presented, effectively duplicating the manuscript text. We have attempted to make the results writing clearer and more specific throughout and hope that this will help readers to follow the analyses.

L165: it could be informative adding to the figure what pair differ significantly (Figure 2B).

L165: Thank you for this suggestion. We have edited this figure caption to more clearly describe the significant differences.